# Evaluating the Efficacy of Probiotics in IBS Treatment Using a Systematic Review of Clinical Trials and Multi-Criteria Decision Analysis

**DOI:** 10.3390/nu14132689

**Published:** 2022-06-28

**Authors:** Cecilia Ceccherini, Sara Daniotti, Claudia Bearzi, Ilaria Re

**Affiliations:** 1Consorzio Italbiotec, Piazza Della Trivulziana 4/A, 20126 Milano, Italy; sara.daniotti@italbiotec.it (S.D.); ilaria.re@italbiotec.it (I.R.); 2Institute of Biomedical Technologies, National Research Council, Via Fratelli Cervi 93, 20054 Segrate, Italy; claudia.bearzi@cnr.it

**Keywords:** microbiota dysbiosis, probiotics, irritable bowel syndrome, nutraceuticals, clinical trials, selective efficacy, multi-criterial decision analysis, PRISMA

## Abstract

The evaluation of probiotics’ efficacy in treating irritable bowel syndrome is supported by an increasing number of clinical studies based on a heterogeneous approach of products tested and the patient cohort involved. Although the role of gut microbiota dysbiosis in IBS pathogenesis and the beneficial contribution of probiotics were demonstrated, a tool to discriminate symptom-specific strains and a personalised medicine protocol are still lacking. Thus, this study employs, for the first time, a method that combines the preferred reporting items for systematic reviews and meta-analysis and multi-criteria decision analysis methods in a structured decision-making tool to analyze the efficacy of probiotic mix, in order to identify the most effective formulation and to discriminate which probiotics are more efficient in treating different symptoms. The PRISMA methodology resulted in a qualitative and quantitative analysis of 104 clinical studies from 2011 to 2021, revealing a prevalence of *Lactobacillus rhamnosus*, *Lactobacillus acidophilus,* and *Bifidobacterium animalis* subsp. *lactis*. MCDA analysis showed that formulations based on *Lactobacillus rhamnosus* and *Lactobacillus acidophilus* have the highest efficacy, especially on quality of life, bloating, and abdominal pain. This methodological approach could become more specific by modelling clinical studies according to the age and gender of patients and probiotic strain.

## 1. Introduction

The correlation between alteration of the intestinal microbiota and the development of inflammatory disorders, such as irritable bowel syndrome (IBS), has benefited an increasing number of studies over the last decade [1,2,3,4]. Beyond the role played by the microbial community in the pathogenesis, these studies have supported the identification of specific probiotic strains in the treatment of IBS symptomatology, such as *Lactobacillus rhamnosus*, *Lactobacillus acidophilus*, and *Bifidobacterium animalis* subsp. *Lactis* [4,5,6]. IBS is a disorder characterized by multifactorial pathogenesis and complex symptoms. Bacterial heterogeneity and concentration of different species affect patients’ symptomatology. The symptoms involve impaired gastrointestinal motility, high visceral hypersensitivity, compromised gut microbiota, and the dysregulation of the microbiome–brain–gut axis (the relationship between the microbiome and the central nervous system), which could be associated with depressive episodes and altered emotional responses [4,7]. IBS is classified into subtypes: IBS with diarrhoea (IBS-D), IBS with constipation (IBS-C), IBS with mixed symptoms of constipation and diarrhoea (IBS-M), and untyped IBS based on recurrent abdominal pain related to defecation or in association with change in stool frequency or shape [3].

*Bifidobacterium*, *Lactobacillus*, and the combination of these two species with *Streptococcus* are the leading probiotic strains preventing, mitigating, or relieving IBS symptoms of visceral tenderness, intestinal permeability, and inflammation. In addition, *Saccharomyces boulardii* improves intestinal flow, *Lactobacillus rhamnosus* reduces abdominal pain, and *B. infantis,* combined with other probiotics, reduces abdominal pain and flatulence [5,8,9]. Analysis of clinical trials suggests that specific combinations of probiotic strains such as *Lactobacillus rhamnosus*, *Lactobacillus acidophilus*, and *Bifidobacterium animalis* subsp. *lactis* could lead to an overall symptom improvement.

Although the benefits of probiotics in treating IBS are proven, personalized medicine protocol and symptom-specific administration guidelines are not available yet. However, the studies comparing clinical trials testing the use of probiotics for the treatment of IBS that are currently found in literature do not start from a systematic literature review, do not focus on evaluating the efficacy of the probiotic with respect to the individual symptom and patient characteristics, nor make use of a statistical prediction method to obtain efficacy comparisons; rather, they focus on generic considerations with respect to the conduct of the trial and the duration of treatment, so there are no structured methodologies to compare the efficacy and discriminate the effect of the probiotic on the symptom [10,11,12].

Efficacy tests of probiotic formulations on mixed patient cohorts result in a partial assessment, due to microbiota variability and its dysbiosis dependence on the subject’s age, gender, and subtype of IBS. Conversely, clinical trials on segmented classes of patients divided by age (child, adult, anxiety), gender, and the IBS subtype (C, D, M) would contribute to a more comprehensive probiotic formulation administration. Such clinical trials would indeed result in an evaluation of probiotics efficacy specific to the characteristics of the microbiota of the considered class (e.g., adult woman with IBS-C; anxious man IBS-D), leading to higher treatment success [3,13,14].

The aim of this study is to take a first step towards a more structured analysis of the effects of probiotics on IBS. Starting with the most studied probiotic products over the last 10 years, the aim is to identify the most effective probiotic formulation for the treatment of IBS and to understand whether the formulations identified in these studies improve IBS symptoms equivalently or have a greater effect on certain symptoms.

Based on these considerations, the present study analyses clinical trials from 2011 to 2021 evaluate the effect of probiotics, prebiotics, and symbiotics administered individually or in strain combinations in treating IBS symptoms.

Based on the preferred reporting items for systematic reviews and meta-analysis (PRISMA) method, a systematic literature review led to the state-of-the-art definition and future perspectives on nutraceutical use [15]. The analysis was completed by the multi-criteria decision analysis (MCDA), comparing the selective efficacy of alternative strains formulation (options) in mitigating IBS symptoms (criteria) [16]. For the first time, the MCDA method was used to analyse the efficacy of probiotic mixtures. Thanks to this method, it was possible to identify the most effective formulation and to discriminate which probiotics are more efficient in treating one symptom than another.

Leveraged in the decision making of different healthcare areas, MCDA helps policy design, resource allocation balancing cost and benefits criteria [17], and investments or decommissioning interventions [18,19]. MCDA is broadly used in clinical settings, such as determining the optimal choice of healthcare infrastructure [20], assessing the quality of long-term care programs, quantifying the effectiveness among treatment options [21], selecting new medical devices in health technology assessment (HTA) [22], and early diagnosing of psychological disorders [23].

The integration of patients’ preferences into the therapeutic decision-making process is supported by HTA-based MCDA [24] for evaluating cure options and screening priorities [25].

In the nutraceutical area [26], MCDA assessed the risk of adverse cardiovascular effects from multi-dietary supplements and the safety of botanical ingredients [25].

Despite these relevant contributions, no MCDA-based studies were addressed until now to assess the efficacy of probiotics, prebiotics, or symbiotics in personalized medicine treatments.

For the first time, this study used MCDA to evaluate the efficacy of three different formulations of probiotic strains in treating IBS, resulting in a symptom-specific efficacy ranking. Based on criteria including quality of life and mitigation of the most relevant symptoms, this approach considers gastroenterologists’ judgements in defining the severity and frequency of IBS symptomatology. The expert assessment was supported by the analytic hierarchy process (AHP) as part of the MCDA to derive paired ratio scales reflecting the relative strength of preferences. This innovative approach is based on quantitative and qualitative data from the literature, thus favouring the standardisation of the procedure.

## 2. Materials and Methods

A dual methodological approach was employed for this study: an initial systematic analysis of clinical trials for IBS from 2011 to 2021 using the PRISMA method, followed by the application of the MCDA approach to identify the most effective probiotic combination for IBS treatment based on the defined criteria (IBS symptoms, quality of life) with the final aim of providing indications to support pharmacological therapy.

### 2.1. PRISMA Method

The PRISMA method, a tool developed by a group of clinical researchers in 2005, supports systematic reviews and meta-analyses consisting of a series of explicit and reproducible steps for identifying, selecting, and evaluating articles relevant to the research topic [15]. It allows the identification of all potentially pertinent studies by querying dedicated databases, selecting eligible ones consistent with the research scope, assessing the risk of bias, and the subsequent extraction and qualitative and quantitative synthesis (meta-analysis) of the results [15]. The PRISMA 2020 Statement is an integral part of the method. This study used the new PRISMA guidelines, “The PRISMA 2020 Statement: an updated guideline for reporting systematic reviews”, published in March 2021 as an updated edition, with new insights from the 2009 version [5,27,28]. The following paragraphs summarise the process and criteria adopted in applying the PRISMA method to assess the qualitative and quantitative trends and perspectives of scientific research on IBS based on prebiotics, probiotics, and symbiotics.

#### Identification of Studies Using Databases and Registries

The study considered scientific publications dealing with clinical studies from 2011 to 2021 testing probiotics, prebiotics, and symbiotics in treating IBS symptoms, excluding those not verifying the efficacy of the strains. Inclusiveness and representativeness of the studies were ensured using three different databases: PubMed, Science Direct, and Cochrane. Input queries, listed in Table 1, were searched for in the scientific publication’s title, abstract, or keywords.

By applying the above criteria, 1385 articles dedicated to the IBS study were identified and manually screened to remove duplicates and publications inconsistent with the research aim. The funnel approach recommended by the PRISMA method for eliminating duplicates by crosschecking the title and abstract gives rise to a unique identification of 536 articles. Eligible articles were subjected to a subsequent screening phase to exclude systematic reviews based on trials and meta-analyses, non-topic trials (even if their title contained keywords from our query), drug-only veterinary trials, and the evaluation of the efficacy of prebiotics and probiotics for conditions other than IBS. Studies in which the tested probiotic strains are not explicitly stated were excluded. Finally, the four-step selection process identifies 104 studies, as summarised in the flow diagram in Figure 1.

The articles resulting from the screening were organized into a database providing the following information for each article:-Year: year of publication, between 2011 and 2021.-Title: title of the article.-First author: first author’s name and primary affiliation.-Country: country and continent where the study was conducted.-Journal: peer-review journal of publication, with related impact factor.-Product: probiotic, prebiotic, and symbiotic under investigation by the clinical trial.-Type of composition: type of composition of each product and their possible combined formulation (probiotic, prebiotic, symbiotic, probiotic mix, prebiotic mix, and symbiotic mix).-Composition: bacterial strain i.e., name(s) of the probiotic(s) and prebiotic(s).

The database, including all the 104 selected publications, is available in Appendix A.

### 2.2. Multi-Criteria Decision Making Analysis—MCDA

The quantitative and qualitative analysis of clinical trial data determines the efficacy of the probiotic formulations most frequently used in IBS treatment according to several criteria, including IBS symptoms and quality of life. In the present study, the MCDA allows the identification of the most effective combined formulation of different probiotic strains (probiotic mix) for IBS treatment. The different alternative strain formulations are defined as options, IBS symptoms are criteria, and finally, parameters used to measure strain performance refers to the average symptom improvement rate [29].

The MCDA methodology applied in this study consists of five steps supported by calculations in R Studio© version 1.4.11.03, following the protocol described by Gatto et al. [30,31].

As the first step in MCDA methodology, clinical trials were analysed to identify the literature’s most suitable formulations (options). A collection of 104 articles led to the identification of 34 studies treating IBS with probiotic mix formulations as a result of the PRISMA analysis. The 34 clinical trials were further analysed to identify the most commonly used combinations of probiotic strains. As a result of this analysis, 9 studies were found using *Lactobacillus acidophilus* in combination with *Lactobacillus rhamnosus* and/or *Bifidobacterium animalis* subsp. *lactis*.

The identified studies were further classified according to the mix compositions, thus identifying the options (i.e., the elements to be compared and ranked in the MCDA) of the MCA:-Lactobacillus rhamnosus, Lactobacillus acidophilus [32,33,34,35,36] (nr. 5)-Bifidobacterium animalis subsp. lactis, Lactobacillus acidophilus [37] (nr. 1)-Lactobacillus rhamnosus, Lactobacillus acidophilus, Bifidobacterium animalis subsp. lactis [38,39,40] (nr. 3)

The second constituting element of the MCDA methodology is a set of criteria representing the parameters used to measure the options’ performance. The present study employs seven criteria to assess the efficacy of the probiotic formulation. The first five criteria correspond to IBS symptoms defined by FDA guidelines, including bloating, abdominal pain, constipation, abdominal cramps, and flatulence; the other two refer to the clinical trials’ quality of life and efficacy [41]. For the five symptom-related criteria, the performance of each option was calculated as the percentage change in improvement after the treatment. The severity of each symptom is measured using visual analogue scale (VAS) scores. The VAS scale corresponds to the visual representation of the pain experienced by the patient and consists of a line 10 cm long, where the left end corresponds to “no pain” (0) and the right end to “worst possible pain” (10) [42]. The patient is asked to draw a mark on the line representing the level of pain experienced. The following formula calculated the improvement value between the first and the last formulation’s administration:(Final score − initial score)/initial score × 100

Quality of life was assessed based on a 34-question survey [43] designed to measure feelings of embarrassment, vulnerability, depression, and the influence of the disease on work, sex life, eating habits, and clothing; in 3 articles out of the total it was stated that there was an improvement without reporting the exact % improvement value, and therefore the values were not included in the calculation of the final average, in the remaining articles it was stated that there was an improvement. Finally, the criterion “effectiveness of the clinical trial” was assessed as “number of clinical trials analysed which were not effective”. MCDA criteria are shown in Table 2.

The second step of MCDA implies the creation of a performance table, reporting options as rows and criteria as columns (Table 4 and Table 5 in results). The values reported in this table represent the performance of the option against each criterion. After calculating the improvement for each item, the values for each symptom were averaged to obtain scores measuring the performance of the strains. The scores, thus representing the performance of each option against each criterion, are derived from the literature analysis and are reported in Table 4 in Section 3.

Following the Gatto et al. [27] protocol, the values of the performance table (Table %) were normalized according to a 0–1 value function that allows easy comparison between data. Considering all the scores related to a specific criterion for different options, this approach assigns endpoints so that 0 is the worst case and 1 is the best outcome. A linear value function proportionally allows the conversion of the values from the natural measurement scale to the 0–1 scale using the two endpoints as reference. Results of the normalization are reported in Table 5 in Section 3.

In this study, criteria from C1 to C6 were maximised: the worst outcome (0) corresponds to the minimum value in the normalized performance matrix, while the best outcome (1) corresponds to the maximum value. C7 is minimized: the worst outcome corresponds to the maximum value while the best outcome corresponds to the minimum value [16]. Calculations were performed on R with the scales and the MCDA package as Gatto et al. [27] described.

The third step in MCDA is the weight definition for each criterion. The analytical hierarchical process (AHP) methodology, developed by Saaty [16], was applied to assign weights to each criterion. Weights to the different criteria are assigned to reflect the relative importance of the weight in the decision. This methodology is based on pair-wise comparisons between the selected criteria, which allow subjective assessments of relative importance (answering the question “how important is criterion A compared to criterion B?”) to be converted into a series of overall scores (or weights). In this study, 9 experts (specialists and former experts in gastroenterology) were interviewed, and their answers were collected in the survey template provided by Goepel [44]. Calculation and weights definition were performed on R software using the AHP survey package as suggested by Gatto et al. [27]. In detail, each expert’s individual preference (or weight) for each criterion is calculated as the eigenvector of each AHP matrix, resulting in the so-called “individual weights”. The calculation of the final weight for each criterion for all decision-makers, also called “aggregate weights”, is based on the arithmetic mean of the individual weights for the same criterion. In addition, the standard deviation between the individual weights was calculated to assess the variability between the individual priorities.

The consistency ratio for the matrix of individual judgements is calculated to measure the consistency of the judgements concerning the pure random judgement. In this study, an acceptable CR was considered <0.1. All individual weights were below the threshold and were therefore considered for further calculation.

As the fourth and last step of the methodology, the global performance of each option is calculated as the weighted average of its score for each criterion. Each performance value (*Sij*) is multiplied by its weight (*wi*), and then the products for the same options are added together to obtain the overall score for that option. Given the total number of criteria (*n*), the equation calculates the total score for an option (*TSi*); the formula for calculation results is as follows [29]:$$TSi=\overset{n}{\underset{j=1}{\sum }}wi*sij$$

The MCDA package of R was used to evaluate each criterion’s total and partial score.

The sensitivity analysis protocol by Gatto et al. [27] was applied to assess probiotic formulations ranking variability according to a ±25% criteria weights variations.

#### Limits of the Study

The restricted number of clinical trials eligible for MCDA analysis (9 out of 104) could affect statistical relevance. Approximations were applied to assess the symptom improvement variation resulting from the administration of the probiotic strain formulations. For instance, approximations were applied when the symptom improvement following treatment with strain formulations was quantified by referring to the number of re-spenders. It was defined as subjects reporting a decrease in symptoms of at least 30% from baseline for at least 50% of the intervention time.

## 3. Results

### 3.1. PRISMA Results

The worldwide scientific literature review dedicated to clinical trials for IBS using probiotics, prebiotics, and symbiotics led to the selection of 104 articles, further analysed according to the publication trend from 2011 to 2021 using a polynomial regression model, as illustrated in Figure 2. A steady trend of scientific production can be attested with an annual average of about 15 articles, reaching its peak in 2020 (nr. 19 articles), an increase of 73% compared to 2011. A marked scientific production can be observed in the 2018–2020 period (nr. 47 articles out of 104).

The scientific production extent, based on the origin country affiliation of the first author, is led by the United Kingdom and Italy with ten articles each, followed by China and Korea (nr. nine each), Iran (nr. seven), USA and India (nr. six each), and France, Denmark and Poland (nr. five each). The geographical distribution of publications is shown in Figure 3, where the top 10 countries account for 69% of world scientific production (nr. 72 of 104 articles).

About half of the scientific production (nr. 54 studies) is attested in peer-reviewed journals in gastrointestinal diseases, translational and clinical basic studies, and nutrition fields, such as *Clinical Trials* (nr. 10 studies, IF 2.4), *Nutrients* (nr. 8 studies, IF 4.5), and *World Journal of Gastroenterology* (nr. 8 studies, IF 3.665). 

Furthermore, clinical trials were classified into three groups according to their formulation composition:-probiotic, tested individually and combined forms of different strains;-prebiotic, tested individually and combined form of different strains;-symbiotic, testing the effect of combined forms of probiotics with prebiotics.

Probiotics are analysed in 65% (nr. 68) of the selected clinical trials, followed by prebiotics (nr. 21) and symbiotics (nr. 15).

Concerning product composition, combined formulations of different probiotic strains were more prevalent than administering a single specific strain. According to the data collected, probiotics and symbiotics in mixed formulations are used in 34 and 15 studies, respectively, out of a total of 104 (approximately 47% of the studies analysed), as shown in Figure 4. Clinical trials using prebiotics, i.e., probiotics in single strains, account for 20% of the analysed studies (nr. 21) with a prevalence of *Lactobacillus* (nr. 14), including *L. acidophilus*, *L. plantarum*, *L. reuteri*, *L. casei*, *L. gasseri*, *L. paracasei*, *L. rhamnosus*, followed by *Bifidobacterium* (nr. 8), *Saccharomyces* (nr. 7), *Bacillus coagulans* (nr. 3), *and Bacterium* (nr. 2).

Finally, a more detailed analysis of the probiotic mix composition was carried out to identify the most recurrent probiotic species in formulations tested on IBS patients. The most recurrent species in the mixed probiotic formulations were *L. acidophilus* (nr. 11), *L. rhamnosus* (nr. 9), *and Bifidobacterium animalis* subsp. *lactis* (nr. 10). Looking at the different combinations between them, *Lactobacillus rhamnosus* represents the most used species, together with *L. acidophilus and Bifidobacterium animalis* subsp. *lactis*, as shown in Table 3.

From all articles selected by the PRISMA method, as described above in Materials and Methods, 9 articles regarding IBS treatment were selected based on frequently studied probiotic strains formulations.

In the selected studies, the efficacy of combined probiotic formulations for IBS treatment is evaluated using randomised, double-blind, placebo-controlled clinical trials for an average duration of 10 weeks, involving an average of 100 subjects of both sexes, aged between 18 and 65/70 years. In 55% of these trials, the variation in microflora composition is analysed by quantitative PCR analysis to observe which strain increases, decreases, or remains stable during treatment and investigate a probiotic–symptom link such as bloating, pain, flatulence, and constipation.

Finally, changes in levels of C-reactive protein and fecal calprotein, both inflammatory markers, were considered in 11% of the trials to highlight how the immune system contributes to the pathogenesis of IBS.

Of the trials, 56% tested the efficacy of formulations of a probiotic mix composed of *Lactobacillus rhamnosus* and *Lactobacillus acidophilus*, 11% of the studies evaluated the efficacy of *Lactobacillus rhamnosus* combined with *Bifidobacterium animalis* subsp. *lactis*, and 33% evaluated the efficacy of the combination of *Lactobacillus rhamnosus*, *Lactobacillus acidophilus,* and *Bifidobacterium animalis* subsp. *lactis* in equivalent amounts.

Overall, 78% of trials showed an average improvement of 55% in symptoms. In 22% of studies, there was no significant enhancement in patients.

### 3.2. MCDA Results

The quantitative analysis of the clinical trials data allowed the implementation of MCDA step 1, i.e., the performance matrix, as shown in Table 4. The first formulation shows the highest values in all criteria (except constipation, scoring 11%). In particular, for the *L. rhamnosus-L. acidophilus* probiotic mix, the criterion “bloating” was associated with a 45% improvement, and the criterion ‘quality of life’ with a 73% increase. The second probiotic formulation (*L. rhamnosus-B.animalis* subsp. *lactis*) improves 30% in almost all criteria, with a 22% improvement in quality of life and zero ineffective studies. The third formulation (mix of three) shows the lowest efficacy values in almost all criteria, except for the “cramps” criterion and the “constipation” criterion registering a 37% and 24% improvement, respectively.

The normalisation of the values, according to a 0–1 scale required by step 2 of the methodology, led to the final performance matrix facilitating the comparison of results, as shown in Table 5.

Nine experts assessed the criteria weights. Among the seven criteria included in the analysis, the highest weight was assigned to C6 (“quality of life”) followed by C2 (“pain”); the lowest weight was assigned to C5 (“constipation”). According to experts, “quality of life” is the most influential criterion in determining the efficacy of a probiotic mix for treating IBS. As described in Figure 5, criterion C1 (“bloating”) has the largest standard deviation, meaning that experts have different opinions about the importance of criterion C1 in assessing the final ranking of IBS treatments. According to the experts‘ evaluation, this discrepancy is due to the close correlation of this criterion with the sex of the patients, as detailed in the discussions. Conversely, criterion C3 (“cramps”) has the lowest standard deviations, meaning that experts agree on its value in assessing the final ranking of IBS treatments.

Final score calculation (Figure 6) allows the ranking of the different probiotics’ formulations, revealing the strengths and properties of each option.

With a final score of 74, the most effective option is the formulation with a predominance of *Lactobacillus rhamnosus* and *Lactobacillus acidophilus* (formulation 1), followed by *Lactobacillus rhamnosus* combined with *Bifidobacterium animalis* subsp. *lactis* (formulation 2) and the mix of three species in equivalent amounts (formulation 3). In particular, the formulation with *Lactobacillus rhamnosus* and *Lactobacillus acidophilus* strongly affects the ‘quality of life’ criteria, contributing 27% to the final score. The graph also shows that formulation 1 of probiotics would positively improve abdominal pain, as well as flatulence and abdominal cramps. These criteria account for the final score of 20%, 9%, and 8%. No contribution to the constipation criterion is shown.

The major contribution to the *Lactobacillus rhamnosus and Bifidobacterium animalis* subsp. *lactis* score comes from the “efficacy” criterion, accounting for 19% of the final score. All the studies analysed for this combination were indeed considered effective. Scoring the lowest for almost all criteria, the mix of the three strains has also the lowest total score with a major contribution from the “cramps” criterion.

The variance of the results by sensitivity analysis tests for each option as ±25% of each criterion varies, as shown in Figure 7. Applying a variation of ±25% does not change the options’ ranking, demonstrating the method’s robustness and, therefore, the data obtained from the clinical trials.

## 4. Discussion

Treatments of IBS-related microbiota dysbiosis are influenced by complex pathogenesis and several external and internal factors such as environment, lifestyle, genetics, and the microbiota itself, whose composition and interaction with the host are essential for health. Probiotics demonstrated efficacy in alleviating the symptoms of various inflammatory diseases [45,46], representing 67% of the clinical trials analysed (nr. 104), with a large population of strains (over fifty recorded) modulated in heterogeneous numbers and percentages. Despite this variety, *Lactobacillus rhamnosus* is the most abundantly studied strain, whose efficacy is mainly assessed in formulations with *Bifidobacterium animalis* subsp. *lactis* in combination with *Lactobacillus acidophilus* and alone (see Section 3.1).

The analysed clinical trials follow the National Institute for Health and Care Excellence (NICE) recommendations on adult IBS in primary care. According to the trials screening showed in Section 3.2, the most effective formulation is characterized by a prevalence of *Lactobacillus rhamnosus* with *Lactobacillus acidophilus* (formulation 1), which affects patients’ quality of life and decreases all IBS symptoms. The clinical studies classified in this group are based on a patient cohort of 82% women with a mean age of 36 years and suffering from IBS-M and IBS-D subtypes. Although comment on the gender incidence is precluded, the formulation based on *Lactobacillus rhamnosus* and *Bifidobacterium animalis* subsp. *lactis* (formulation 2) could be recommended to treat IBS-C in adult patients. Finally, the formulation of *Lactobacillus rhamnosus*, *Lactobacillus acidophilus,* and *Bifidobacterium animalis* subsp. *lactis* in equivalent amounts (formulation 3), despite being the least effective of the three options tested, could be administrated to IBS-D in adult males as demonstrated by the efficacy rate of the clinical studies on patients’ cohort consisting of 65% males with a mean age of 42 years. The efficiency of formulations 1 and 3 has been study in a limited number of studies; therefore, it would be desirable to increase trials and modelling studies on pain and quality of life, a priority confirmed by the external experts recruited.

In light of the MCDA results, new considerations arise regarding the design of the protocols. Randomised double-blind clinical trials include lyophilized and encapsulated probiotic formulations administrated for a cycle ranging from 12 weeks (33%) to 4, 6, and 8 weeks (67%). The patient cohort comprises 100 subjects on average, with predominant female gender (specified in 83% of the trials) aged 18 to 70 years (ethnicity of the subjects is specified in only 16% of the studies). Concerning IBS subtypes, there is an inhomogeneous distribution between IBS-D (33% of the studies), IBS-C (16%), and in the remaining trials (50%), the same treatment was tested for IBS-D and IBS-M. All studies include an HR-QOL and IBS-QOL quality of life questionnaire assessing symptoms with the VAS scale, ranging from 0 to 10. C-reactive protein and calprotectin were analysed in 11% of trials. Technical analyses include DNA extraction, assessment of the faecal microbiome by quantitative PCR, and 16s RNA analysis required to understand how the microbiota changes with treatment. Protocols’ heterogeneity for administration duration, cohort size, gender, and IBS subtypes affect the pair-wise trials comparison and results’ statistical significance. Having a single subtype of IBS protocol with a standard (minimum size) number of patients could contribute to avoiding protocol bias.

The clinical studies analysed in the present study highlight a gender imbalance in patients’ recruitment: the patient cohorts included in these clinical trials often included more women than men. This may outline a higher prevalence of IBS in women than in men. Furthermore, it appears that women have a different perception of the symptoms. According to judgement of experts involved in this study, ‘bloating’—a symptom mitigated by formulation nr.1—is highly stressful and causes significant discomfort for women, although it may be irrelevant for men. Experts also affirmed that women affected by IBS feel more fatigue, depression, anxiety, and, in general, experience a lower quality of life than men affected by the same disorder. In addition, other chronic pain disorders such as fibromyalgia, chronic fatigue syndrome, chronic pelvic pain, and migraine are more frequent in women with IBS, suggesting an association between their symptoms and hormonal status. These considerations are in line with findings of several previous studies: IBS is a disorder that predominantly affects women with an odds ratio of 1.67, with symptoms that are more severe for them in comparison to men affected by IBS. The reasons for this gender imbalance can be traced back to biological and psychosocial factors related to gender [47,48,49].

This different perception resulted in a large standard deviation in criteria weighting. Clinical protocols considering gender balance issues in evaluating symptoms should be designed (cohorts homogeneous by gender and sub-type of IBS), as gender bias can affect the evaluation of probiotic formulation efficacy.

Although clinical protocols must consider the patient’s subjective evaluations of symptom improvement, these should be included in a wide range of measurement scale criteria. Clinical trials should assess treatment efficacy based on quantifiable, objective, and comparable markers specific to inflammation, which is involved in the pathogenesis of IBS and caused by microbiota dysbiosis. For example, the analysis of C-reactive protein and faecal calprotectin, an antimicrobial protein found in neutrophils released into the intestinal lumen in case of inflammatory processes in the intestine, can be detected in faeces, could be adopted as the main tests for efficacy assessment. These, accompanied by symptom assessment using the VAS scale, IBS-SSS, and the HR-QOL quality of life assessment, could provide a complete picture of the probiotic efficacy on the patient. In addition, it would be necessary to define better each element of symptomatology, thus increasing the reliability of the subjects’ responses. Flatulence is not synonymous with discomfort; abdominal pain should be distinguished from abdominal cramps. Constipation and abdominal pain should be distinguished from abdominal cramps, and constipation should not be grouped with stool shape. It would also be advisable to include a microbiome analysis with RNAs in each study to see how the microbiota varies, which helps the understanding of the effect of the product on specific classes of bacteria.

Finally, the study results show that the efficacy of formulations is specific not only to the gender of the patient, but also to a particular age group and the particular IBS subtype. People of different ages indeed have different microbial compositions and different subtypes of IBS corresponding to different microbiota dysbiosis. Therefore, recommending the same formulation without considering gender and age differences may not be the best solution, considering how varied the condition is. Studies should therefore be conducted by creating cohorts of patients in statistically significant numbers and divided by age and sex, testing a single probiotic formulation on a single subtype of IBS (e.g., women—25–35 years of age), which may lead to better efficacy results and more realistic considerations.

## 5. Conclusions

The consumption of probiotics grew substantially within the last decade, driven by public awareness of ‘good bacteria’ as a health ally. Probiotic strains such as *Lactobacillus rhamnosus*, *Bifidobacterium animalis* subps. *lactis,* and *Lactobacillus acidophilus* are helpful in treating chronic and debilitating inflammatory diseases such as IBS, characterized by complex gastrointestinal tract symptoms. Thanks to the MCDA approach, firstly applied to the analysis of nutraceuticals for clinical applications, an objective comparison of the beneficial effects of probiotic formulations in improving IBS symptoms was possible. The available results highlight the importance of further studying the promising strains for treating IBS disease. Furthermore, the study results could support standardising therapeutic indications according to the patient’s age, sex, and prevalent symptoms. Finally, extending this PRISMA-MCDA combined methodology to other diseases could be an ex-ante tool to obtain predictive data for targeted trials.

## Figures and Tables

**Figure 1 nutrients-14-02689-f001:**
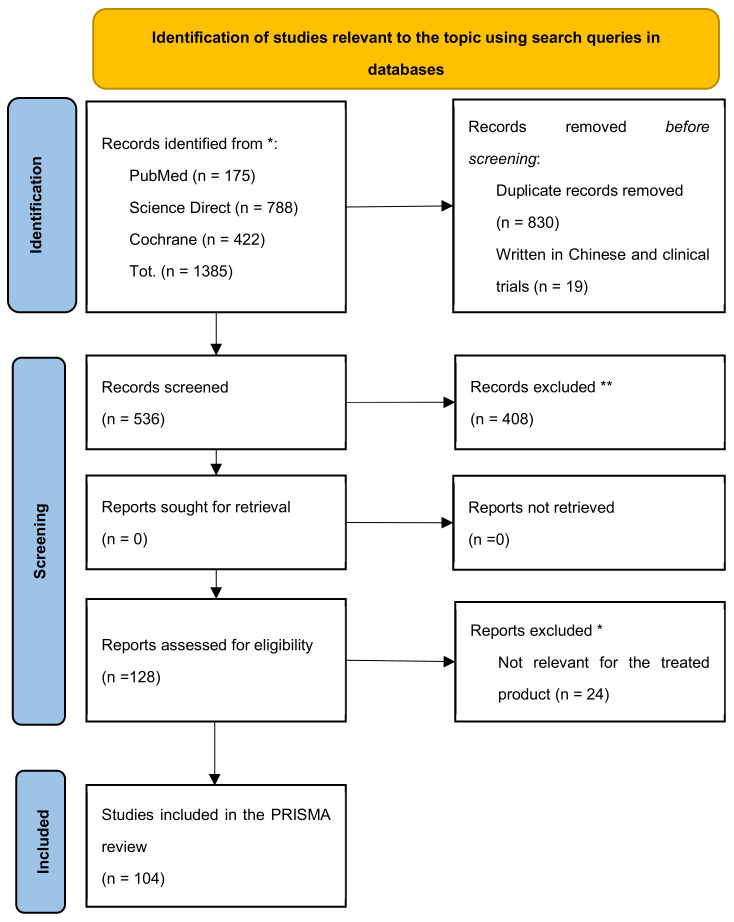
PRISMA 2020 diagram to support systematic analysis of clinical trials in IBS. * query strings for the systematic research in the available databases are detailed above; ** exclusion criteria are detailed in the paragraph above.

**Figure 2 nutrients-14-02689-f002:**
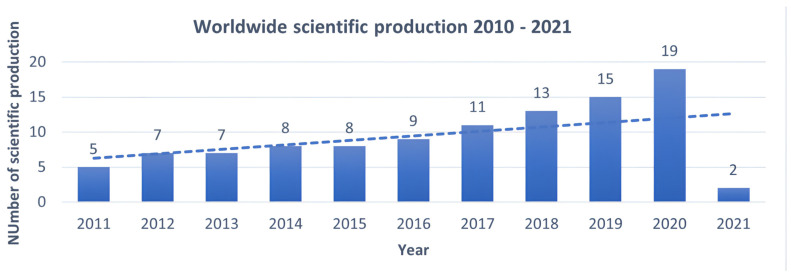
Evolution of worldwide scientific production 2011 – 2021 dedicated to evaluating the efficacy of probiotics, prebiotics, and symbiotics in treating IBS. The *Y*-axis shows the number of publications, the *X*-axis the years in which they were published. This means that interest in the topic is increasing over the years.

**Figure 3 nutrients-14-02689-f003:**
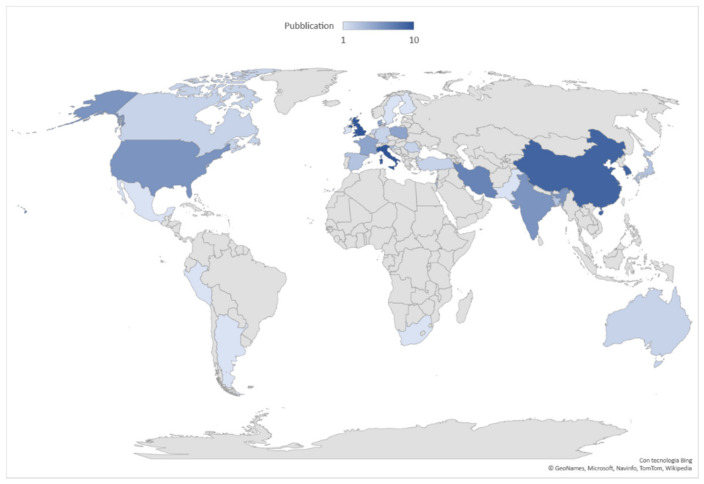
Geographical distribution of worldwide scientific production 2011–2021. Data are broken down into 31 countries. In dark blue the countries in which the most articles on the subject were published.

**Figure 4 nutrients-14-02689-f004:**
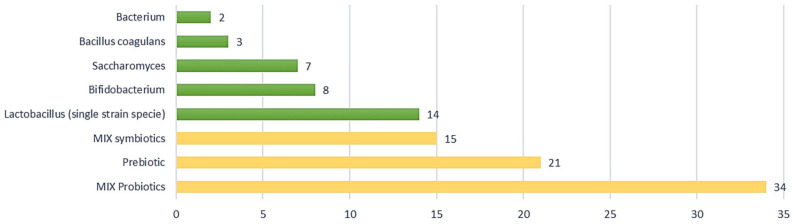
Type of composition in IBS clinical trials; 67% of studies (in yellow) examined combinations of probiotics (MIX probiotic nr. 34), prebiotics (nr. 21), and combinations of symbiotics (MIX symbiotic nr. 15).

**Figure 5 nutrients-14-02689-f005:**
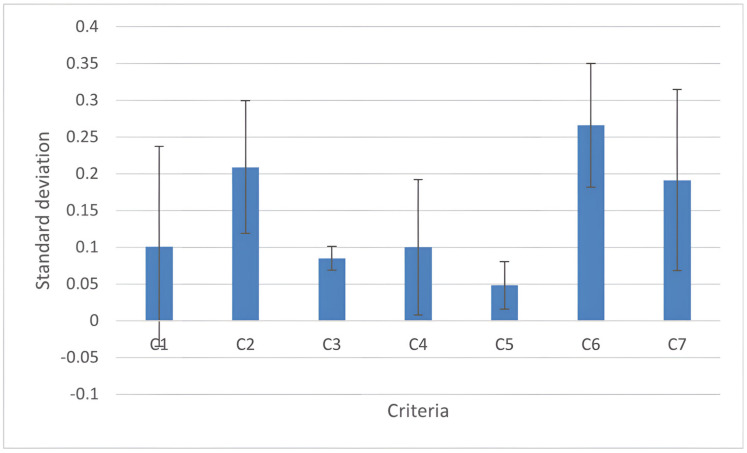
Standard deviation of individual criteria values: the criterion C1 (bloating) has the largest standard deviation; C2 (pain); C3 (cramps); C4 (flatulence); C5 (constipation); C6 (quality of life); and C7 (efficacy). The standard deviation refers to the opinion about the importance of criteria in assessing the final ranking of IBS treatments. The values that correspond to the average of the doctors’ ratings for each criterion are shown in blue. The error bar indicates the standard deviation referring to that opinion.

**Figure 6 nutrients-14-02689-f006:**
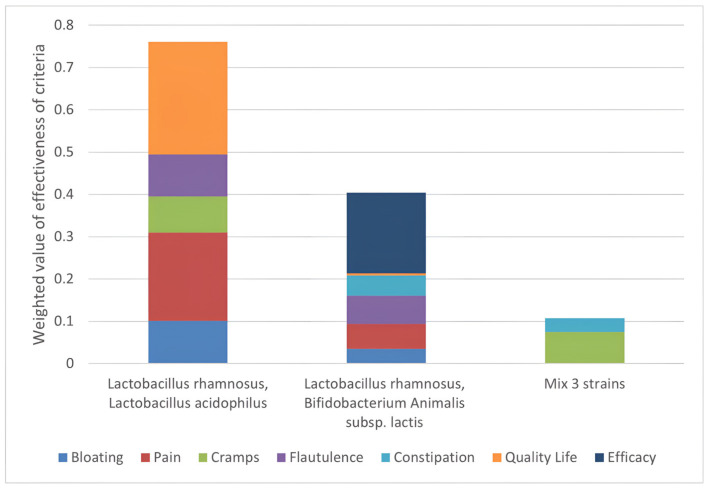
Strengths of each option. Ranking of the effectiveness of the options assessed against the established criteria and weighted to obtain a priority scale. On the Y-axis is the score that each mix has obtained: it is the result of the % improvement of each symptom, weighted by the medical evaluation performed. It is shown that the symptoms “quality of life” and “abdominal pain” are the symptoms where most action is needed and on which formulation 1 has the greatest positive effect.

**Figure 7 nutrients-14-02689-f007:**
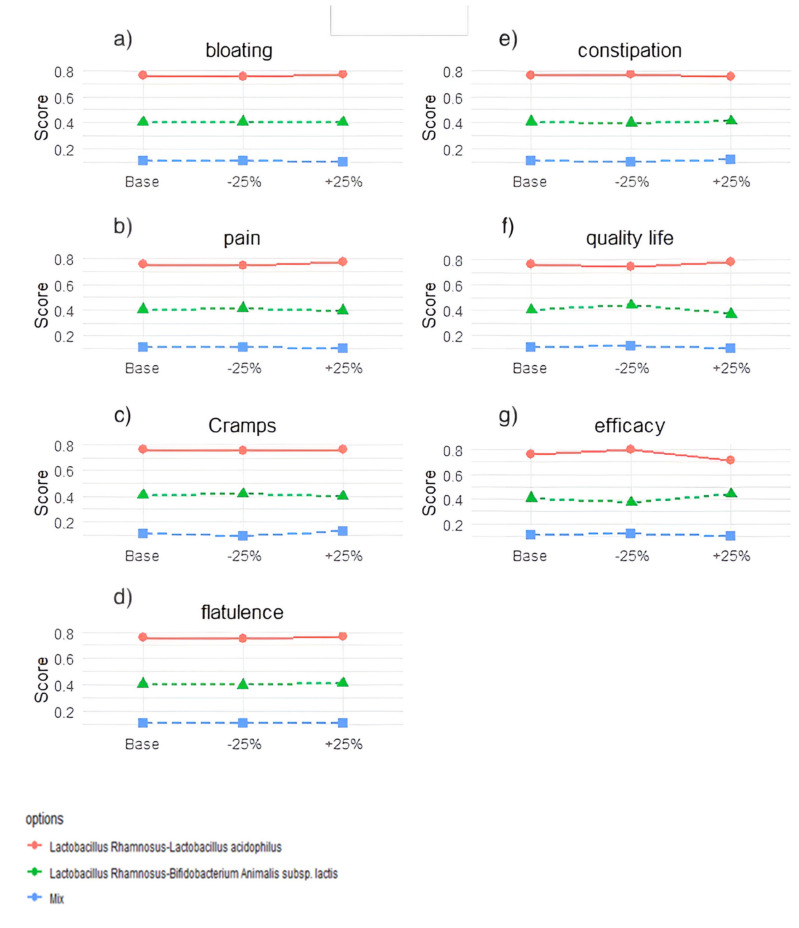
Results of the sensitivity analysis. Each graph refers to one criterion and shows how the total score for each probiotic mix changes for a 25% increase or decrease in the weight of that criterion compared to the baseline scenario. (**a**) C1 = bloating; (**b**) C2 = pain; (**c**) C3 = cramps; (**d**) C4 = flatulence; (**e**) C5 = constipation; (**f**) C6 = Quality of life; (**g**) C7 = efficacy.

**Table 1 nutrients-14-02689-t001:** Query strings for the systematic search of articles. We reported the number of articles founded in different databases and the query used for search publications.

Query	Science Direct	PubMed	Cochrane
IBS clinical trials probiotics	236	76	148
Clinical trials probiotics IBS	236	76	186
IBS clinical trials prebiotics	111	10	33
Clinical trials prebiotics IBS	111	10	41
Prebiotics probiotics IBS clinical trials	94	3	14

**Table 2 nutrients-14-02689-t002:** IBS common criteria. Seven criteria were employed to assess formulation efficacy; criteria match the symptoms of IBS. This table shows the measurement used for each criterion, the score, and the evaluation type (related to the MCDA method).

Criterium	Criterium Title	Measurement	Score Type	Evaluation Type
C1	Bloating	% Value of the improvement	VAS (0–10)	Maximization
C2	Pain	% Value of the improvement	VAS (0–10)	Maximization
C3	Cramps	% Value of the improvement	VAS (0–10)	Maximization
C4	Flatulence	% Value of the improvement	VAS (0–10)	Maximization
C5	Constipation	% Value of the improvement	VAS (0–10)	Maximization
C6	Quality Life	% Value of the improvement	Q-score (100–300)	Maximization
C7	Efficacy	Number of trials resulting not effective	Yes/No	Minimized

**Table 3 nutrients-14-02689-t003:** IBS prevalent probiotic combinations. Out of 11 combinations with *Lactobacillus acidophilus*, 7 are characterised by the presence of *Lactobacillus rhamnosus*.

IBS Probiotic Combinations	*Lactobacillus acidophilus*	*Lactobacillus rhamnosus*	*Bifidobacterium animalis* subsp. *lactis*
*Lactobacillus acidophilus*	11	7	5
*Lactobacillus rhamnosus*	7	9	6
*Bifidobacterium animalis* subsp. *lactis*	5	6	10

**Table 4 nutrients-14-02689-t004:** Performance matrix for MCDA. In clinical trials, patients evaluate their symptoms at week zero and at the final week (after taking the product) on the VAS scale. The values shown here are the % improvement of symptoms as described in Section 2.2 multi-criteria decision-making analysis—MCDA.

	Bloating	pain	Cramps	Flatulence	Constipation	Quality Life	Efficacy
*L. rhamnosus-L.acidophilus*	45	48	38	38	11	73	1
*L.rhamnosus-B.animalis* subsp. *lactis*	30	30	30	30	30	22	0
Mix 3 species	22	23	37	14	24	21	1

**Table 5 nutrients-14-02689-t005:** MCDA normalized matrix. The options were rated on a 0–1 scale so that the differences were consistent within each criterion, making the data easily comparable. The present study applies a 0–1 scale to the values according to a linear value function that assigns endpoints such that 0 is the worst case and 1 is the best outcome.

	Bloating	Pain	Cramps	Flatulence	Constipation	Quality Life	Efficacy
*L.rhamnosus-L.acidophilus*	1	1	1	1	0	1	0
*L.rhamnosus-B.animalis* subsp. *lactis*	0.34	0.28	0	0.67	1	0.02	1
Mix 3 species	0	0	0.86	0	0.68	0	0

## Data Availability

Data employed and generated by this study are all available in the material and methods section. The R-code presented and employed in this study is available in Gatto et al. (DOI: https://doi.org/10.3390/su132111709 accessed on 6 June 2022). The database, including the 104 selected publications, is available in Appendix A.

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
