# Peer review of "Evaluating the Efficacy of Probiotics in IBS Treatment Using a Systematic Review of Clinical Trials and Multi-Criteria Decision Analysis"

_nutrients, 2022, doi:10.3390/nu14132689_

Round 1
Reviewer 1 Report
In this study, the authors combined the Preferred Reporting Items for Systematic Reviews and Meta-Analysis (PRISMA) and the Multi-Criteria Decision Analysis (MCDA) methods to analyzed and screed the most efficient combination of bacterial strains in treating irritable bowel syndrome symptoms through testing probiotics, prebiotics and symbiotics. Their analysis showed that formulations based on Lactobacillus rhamnosus and Lactobacillus acidophilus have the highest efficiency with positive effects on quality of life, bloating and abdominal pain. The article is well written, didactical and has methodologies.
I have mineral comments to improve the manuscript:
1. In Figure 2 and figure 6, the legend of x and y axis should be labeled.
2. Please check whether a full stop should be in line 449.
Reviewer 2 Report
Ceccherini and colleagues present an interesting systematic review combined the Preferred Reporting Items for Systematic Reviews and 17 Meta-Analysis (PRISMA) and the Multi-Criteria Decision Analysis (MCDA) methods.
The methodology used is accurate, and the exclusion criteria for obtaining own data to compare the efficacy of probiotics in IBS preclinical trials have been appropriate.
The document was written in a reasonable manner, but the results and discussion sections must be improved. Certain data are unspecified, for example the generality of the strains used. Only the genus and species are mentioned, it can be assumed that they are the same strains for all the pre-clinical studies cited, Please specify the strains used. If the strains are stored in international collections, please indicate their name.
The discussion should be improved, an appropriate contrast of ideas with other reports (as well as cited them) should be considered. For example, they deal with the efficacy of probiotics in women and men, mentioned only the correct use of statistical tools, but never make inferences about other approaches. The discussion section should be further developed.
All Figures and Tables should have a short explanatory title and caption. Provide sufficient information in the tables and figures, readers do not need to read the text to understand it.
Line 32: Is the IBS considered as a disease?
Line 36: subps no italics
Line 39: is not clear this part: gut-brain axis interruption
Line 41: This sentence should to be moved Bacterial heterogeneity and concentration of different species affect patients' symptomatology
Line 47, preventing or maintaining?
Line 53: please provide more information about the methodology employed to obtained the 53%,
Line 296: Check please the Y-axis, MIX Probiotic instead MIX Probiotici?
Line 302-305: when authors refer as "the most recurrent probiotic strains were" thats means that 11 studies used the same strain or is about the specie strains?, if is possible, please include the strain used in the studies.
Table 3: Name of microorganisms in italics.
Line 354: please, explains which symptoms.
Table 4: missing information, the information provided is not enough to understand the table for itself.
Table 4 and Table 5 could be reedited and crate only 1 table
Table 5: symptoms names in capital letter or not? Please homogenize
Line 343-355, this part is not clear, and figure 5 must to be improved, please for the figure legend description respect the order C1 to C7.
Line 423-429, develop more about the gender difference in symptoms.
Reviewer 3 Report
The subject frame of the work is well constructed. So, in this respect and this article should be contributed to present research.
1. There are several typographical mistakes as well in whole manuscript. Therefore, the author’s thoroughly careful check the language and typo mistake to minimize the error.
2. The abstract should be beginning with a sentence about the background of concept and the aims as well as novelty of study should be mentions. What exactly is the novelty of this study? The abstract is poorly written and should be improved. Abbreviations must be avoided in abstract. Parenthesis should be avoided in abstract - this is poor writing. Please improve.
3. Introduction; Check and format the citations in the whole manuscript. Also, Appropriate references must be provided to explained the background, what is already done and why this study carried out. Other vise the novelty of this research is still poorly presented. This is important especially for the high IF journals. The scientific style should be used. What exactly is the aim of this work? Hypothesis statement is missing in the introduction section.
4. Results and discussion; General remark to the discussion - In my opinion, the discussion provided by Authors is difficult to follow and verify due missing critical details in the methodology section. Due to poorly described material and poorly presented methods, I am not able to follow and properly review the discussion.
5. All figures are of poor technical quality and not suitable for publication, especially in a high reputed journal. Font size and kind is too small and must be unified in all figures. Small writings are unreadable. All figures must be self-explanatory. Axis titles are poorly presented or absent. Units are missing. Are the data presented in figures significantly different? At least error bars should be shown.
6. I suggest first time write full name rather than abbreviation; revise throughout in manuscript
